# Cadaveric Adipose-Derived Stem Cells for Regenerative Medicine and Research

**DOI:** 10.3390/ijms242115696

**Published:** 2023-10-28

**Authors:** Lara Milián, Pilar Molina, María Oliver-Ferrándiz, Carlos Fernández-Sellers, Ana Monzó, Rafael Sánchez-Sánchez, Aitana Braza-Boils, Manuel Mata, Esther Zorio

**Affiliations:** 1Department of Pathology, Faculty of Medicine and Dentistry, Universitat de València, 46010 Valencia, Spain; lara.milian@uv.es (L.M.);; 2INCLIVA Biomedical Research Institute, 46010 Valencia, Spain; 3Department of Pathology, Instituto de Medicina Legal y Ciencias Forenses, 46010 Valencia, Spain; pilar.molina@uv.es (P.M.); fernandez_carsel@gva.es (C.F.-S.); monzo_anabla@gva.es (A.M.); 4CAFAMUSME Research Group, Instituto de Investigación Sanitaria La Fe, 46026 Valencia, Spain; rafa_sanchez@iislafe.es (R.S.-S.); aitana_braza@iislafe.es (A.B.-B.); zorio_est@gva.es (E.Z.); 5CIBERCV, Center for Biomedical Network Research on Cardiovascular Diseases, 28015 Madrid, Spain; 6Inherited Cardiac Diseases Unit, Cardiology Department, Hospital Universitario y Politécnico La Fe, 46026 Valencia, Spain; 7Department of Medicine, Faculty of Medicine and Dentistry, Universitat de València, 46010 Valencia, Spain

**Keywords:** adipose-derived stem cells, mesenchymal cell, cartilage regeneration, bone regeneration, tissue engineering, in vitro disease models

## Abstract

Advances in regenerative medicine have enabled the search for new solutions to current health problems in so far unexplored fields. Thus, we focused on cadaveric subcutaneous fat as a promising source of adipose-derived stem cells (ADSCs) that have potential to differentiate into different cell lines. With this aim, we isolated and characterized ADSCs from cadaveric samples with a postmortem interval ranging from 30 to 55 h and evaluated their ability to differentiate into chondrocytes or osteocytes. A commercial ADSC line was used as reference. Morphological and protein expression analyses were used to confirm the final stage of differentiation. Eight out of fourteen samples from patients were suitable to complete the whole protocol. Cadaveric ADSCs exhibited features of stem cells based upon several markers: CD29 (84.49 ± 14.07%), CD105 (94.38 ± 2.09%), and CD44 (99.77 ± 0.32%). The multiparametric assessment of differentiation confirmed the generation of stable lines of chondrocytes and osteocytes. In conclusion, we provide evidence supporting the feasibility of obtaining viable postmortem human subcutaneous fat ADSCs with potential application in tissue engineering and research fields.

## 1. Introduction

In recent years, regenerative medicine has received a great boost thanks to the optimization of mesenchymal cell isolation and culture protocols. The ethical and legal problems derived from the use of embryonic stem cells have prompted different research groups to develop efficient methodologies to find an alternative, such as the use of stem cells from adult tissues. Thus, the search for the optimal type of adult stem cell for each purpose has just begun, focusing on different sources including bone marrow, dental pulp, and many others [1]. Above all, we highlight the potential role in regenerative medicine of adipose-derived stem cells (ADSCs), probably the most popular of the mesenchymal stem cells (MSCs), given the great advantages with regard to their high potency and mitotic capacity [2].

Mesenchymal stem cells are defined as adherent cells expressing CD10, CD73, and CD90, but not CD45, CD34, CD14, CD11b, and CD79, which are able to differentiate in vitro into osteoblastic, adipogenic, and chondrogenic cells [3,4,5]. Although MSCs were initially described in bone marrow, it is now commonly thought that MSCs are ubiquitous and equivalent in their differentiation properties [5].

Although ADSCs were isolated for the first time in 2001 from lipoaspirates obtained during cosmetic surgeries, they can be also isolated from any adventitia or serosa with recently optimized protocols based on small biopsies of adipose tissue [6,7]. Initially, this cell population was defined as preadipocytes; however, they exhibited histological characteristics of multipotent cells and their ability to differentiate into osteoblasts and chondroblasts was soon proved [6,8]. They were additionally differentiated into skeletal muscle and a wide range of different cell types, such as glial cells, nerves, endothelial cells, or even hepatocytes, among others, leaving no doubt about their pluripotential capacity in regenerative medicine and their appealing use in different research fields [9,10,11,12,13,14].

Tissue engineering evolved from the field of biomaterial development and refers to the practice of combining scaffolds, cells, and biologically active molecules in order to obtain functional tissues. Originally, the goal was to assemble functional constructs to restore, maintain, or improve damaged tissues or even whole organs [15]. Noteworthy, the knowledge derived from the manufacture of artificial tissues and organs opened new lines of action in biomedicine, namely the generation of tissues from other cell types of the same subject in an autologous manner, thus making it possible to avoid the use of immunosuppressants. In this sense, ADSCs represent a widely used source of tissues for articular cartilage regeneration during trauma surgery [16].

Additionally, tissue engineering in research fields entails an equally promising future. In fact, it allows the generation of in vitro models for the study of different pathologies, ranging from complex diseases, such as cancer and Alzheimer’s disease, or even Mendelian genetic conditions. Both pathophysiology research and response to pharmacological challenges can be addressed with these models, sometimes more accurately than with other pre-existing in vitro or in vivo models [17]. When obtained from affected patients, this approach truly represents an example of the so-called personalized medicine [18]. In certain scenarios, such as sudden death caused by inherited cardiac diseases, obtaining cultures from deceased donors may open new avenues to better understanding the underlying lethal genetic condition and to explore new treatment options thanks to tissue engineering protocols, thus improving the management not only of at-risk relatives but also of patients in general.

A few reports of successful adult MSC isolation from animal and human postmortem tissues were reported with variable postmortem interval limits ranging from <24 h to 5 years, depending on the tissue type and storage conditions [19,20,21,22,23,24]. Nevertheless, the optimization of protocols for ADSC isolation and characterization from these donors remains poorly studied even though it could positively impact the worldwide problem of shortage of donors and, at the same time, offer valuable cultures for research with the aim of improving daily clinical practice in certain settings.

In this work, we optimized a protocol for the isolation and culture of ADSCs from cadaveric donors. We characterized the isolated cells according to the expression of stem cell markers and we evaluated their potential to differentiate into cartilage and bone cells.

## 2. Results

### 2.1. ADSC Isolation and Characterization

In 8 out of the 14 donors, mesenchymal cells could be isolated and were further processed. The remaining six discarded samples were contaminated or had definitely stopped their growth. Thus, 57.14% of the initial samples successfully continued the protocol.

Once expanded and before any cell differentiation protocol was carried out, cells were characterized using flow cytometry. The percentage of isolated cells positive for each of the considered markers was as follows: CD29 (84.49 ± 14.07% of cells), for CD146 (16.63 ± 18.45%), for CD105 (94.38 ± 2.09%), for CD44 (99.77 ± 0.32%), for CD45 (1.01 ± 1.25%), for CD31 (0.62 ± 0.14%), and for STRO (36.14 ± 55.20%). Representative cytometry plots of one of the samples included in the study are shown (Figure 1A). Additionally, cell morphology was evaluated using phase contrast microscopy confirming the characteristic features of mesenchymal cells 1 week after having been seeded (Figure 1B).

### 2.2. Evaluation of Chondrogenic Differentiation Capacity of ADSCs

Next, aiming to evaluate the potentiality of ADSCs, we induced the differentiation into chondrogenic cells. To do this, we cultured the ADSCs with a chondrogenic induction medium for 4 weeks. The first finding we observed in the ADSCs was a shift of cellular arrangement from a cell monolayer to a stellate morphology (Figure 2A). This change correlated with a marked increase in the expression of type II collagen and aggrecan in comparison with the original expression level of these proteins detected before starting the differentiation process (Figure 2B).

Irrespective of their origin, differentiated cells from patients and the Lonza cell line tended to detach from their neighbors and showed a more mature cytoplasm represented by the presence of bundles of actin fibers specifically located at the junction sites. Protein changes were confirmed using RT-qPCR in the patients’ and the Lonza cell line cultures, given that a significant increase in the expression of *COL2A1* and *ACAN* was observed in all the experiments after 1 week with chondrogenic medium (Figure 2C). We did not find any significant difference in the yield of the chondrogenic differentiation protocol between ADSCs isolated from patients and the Lonza ADSCs.

### 2.3. Evaluation of Osteogenic Differentiation Capacity of ADSCs

Once the differentiation potential to chondrogenic lineages had been confirmed, we addressed the issue of whether ADSCs could also commit to osteogenic lineage and differentiate into mature osteoblasts. To this end, we cultured them in an osteogenic induction medium for 2 weeks. The cells exposed to the differentiation medium acquired an elongated morphology and grew, covering the entire culture surface. Under the influence of the differentiation medium, cells exhibited a more mature cytoplasm, and cell junctions appeared (Figure 3A). These morphological changes were studied in detail using the fluorescent staining of F-actin, showing the concentration of actin fiber bundles at the binding sites (Figure 3B). The cells in the proliferation medium showed no expression of osteocalcin or alkaline phosphatase and only a residual expression of type I collagen, whereas the osteogenic induction medium provoked a significant increase in the expression of all these proteins when assessed using immunofluorescence (Figure 3B) and with RT-qPCR (Figure 3C). Finally, we did not find any significant difference in the yield of the osteogenic differentiation protocol between ADSCs isolated from patients and the Lonza ADSCs.

## 3. Discussion

The use of MSC from cadaveric donors opens up a world of possibilities, not only as a source of tissues for regenerative medicine (therapeutic organoids), but also as a very valuable tool for the study of the pathophysiological pathways and treatment options for Mendelian diseases in the era of personalized medicine. In this work, we isolated mesenchymal cells from subcutaneous fat biopsies of deceased people with postmortem intervals of up to 55 h. These cells met the main requirements to be considered MSCs [3]. On the one hand, they exhibit firm adhesion properties. On the other hand, they express a profile of markers compatible with MSCs, being positive for CD29, CD105, and CD44 and negative for CD45 and CD31. In the case of CD146 and STRO, we detected a minority population close to 17% and 37%, respectively, of positive cells. The MSC expression of CD146 is heterogeneous and may depend on the tissue type and the molecular environment. Indeed, hypoxia is related to down-regulation in the expression of CD146. This O_2_-regulation was associated with the localization of MSCs so that CD146+ reticular cells were located in the O2-enriched vicinity of vessels, while cells CD146− were close to the bone surface [25]. Subpopulations of CD146+ and CD146− cells were detected in bone marrow MSCs so that cells with a high expression of CD146 were committed towards vascular smooth muscle cell lineage or pericytes [26,27]. Highly expressing CD146 subpopulations were also described in ADSCs, being considered markers of pluripotency [27]. STRO-1+ MSCs possess many of the hallmarks of multipotent stem cells, including high telomerase activity, undifferentiated phenotype, as well as multi-lineage differentiation potential [28]. High STRO-1 MSCs were found to secrete higher levels of paracrine factors that support cardiac muscle cell proliferation and angiogenesis [29]. As expected, different subpopulations of MSCs were isolated from the patients’ samples. This statement provides the rationale to suggest that filtering the selection for a given profile of cells could be useful to increase the yield of the differentiation process for specific cell lineages. Undoubtedly, further research is needed to explore this hypothesis.

Once the isolated progenitors were characterized as ADSCs, we aimed to test their ability to differentiate into different cell types. To do this, we cultured the cells in a chondrogenic and osteogenic induction medium. The differentiation of ADSCs into chondrocytes is widely supported by the literature [30]. The chondral differentiation medium used consisted of a high glucose DMEM medium enriched with insulin-transferrin-sodium selenite medium supplement, ascorbic acid, TGF-β1, and 1% heat-inactivated FBS. This induction medium has already proven its capacity for triggering chondrogenic differentiation [31,32,33]. Cells cultured in a differentiation medium were compared with controls (cells cultured with a proliferation medium) to conclude that the first morphological change suggesting chondral induction was evidenced with F-actin expression analyses. The cells became less elongated and detached from their neighbors; changes that can be considered mild early markers of chondrogenic induction. We acknowledge that these changes could have been more pronounced in 3D cell-culture experiments, such as those reported with hydrogels or microtissues [31,32]. Different markers were used to study the differentiation of MSCs to chondrocytes, including extracellular matrix proteins (type I collagen, type II, aggrecan, type X collagen), matrix metalloproteinases (MMP9, MMP13), tissue inhibitor proteins (TIMP), nuclear factors (SOX proteins, RUNX1), biochemicals (vascular endothelial factors), etc. [33]. In this work, we did not intend to establish the optimal conditions for chondrocyte differentiation. Instead, we just aimed to verify that the isolated MSCs had the potential to differentiate into chondrocytes, thus a 2D culture system was considered appropriate. As markers of chondrogenic differentiation, we chose type II collagen and aggrecan, given that they are the most characteristic proteins of the chondral matrix, and are not expressed by undifferentiated MSCs. These markers were analyzed using immunofluorescence and RT-qPCR. Both techniques demonstrated that the differentiated cultures expressed high amounts of both protein and messenger RNA (mRNA) markers, in agreement with a chondral phenotype acquisition. We did not find any significant difference between the MSCs isolated from cadaveric donors compared to those from Lonza’s cell line, which demonstrates an equivalent potency irrespective of the cell’s origin.

To explore the multipotentiality of the isolated MSCs, we tested their ability to further differentiate into osteocytes. Again, we were not interested in obtaining the optimal differentiation conditions, only in studying whether induction was possible or not. For this reason, we used a simple 2D culture system and focused on the cellular morphological evolution, as well as the expression of type I collagen, osteocalcin, and alkaline phosphatase induced by the differentiation medium. This medium consisted of αMEM supplemented with L-glutamine, glycerol-2-phosphate, ascorbic acid, and dexamethasone. We also added BMP2 based on our previous work reporting the synergistic effects of this molecule in the induction of the differentiation process into osteocytes [34]. We found a significant increase in the expression of mRNA and the protein levels of the target markers, namely collagen type I, osteocalcin, and alkaline phosphatase, paralleling changes in the morphology of the cultures. Indeed, cells acquired a less elongated and stellate morphology, and spaces appeared between them. Additionally, a greater densification of the F-actin filaments was observed, especially in the periphery of the cell extensions, a characteristic feature of cell unions supporting the acquisition of an osteocyte phenotype [35,36]. Again, we did not find significant differences in the differentiation process of the MSCs isolated from patients and those supplied by Lonza’s cell line, reinforcing the similar potency of both cell sources.

Our primary objective was to assess whether MSCs could be isolated from postmortem subcutaneous fat and then differentiated into chondrocytes and osteocytes. If achieved, it would bring closer the promising possibility of being able to obtain engineered tissues to treat patients and expand the knowledge of the pathophysiological mechanisms underlying diseases such as genetic cardiomyopathies, osteoarthritis, osteogenesis imperfecta, periimplantitis, or osteonecrosis, for instance. It could also provide a source of cells to generate pharmacological models to test drugs or the performance of new biomaterials in the regeneration of different tissues. Furthermore, our work could even be useful in understanding the effects of chronically administered drugs on certain tissues, such as in the setting of the long-term administration of bisphosphonates to treat osteonecrosis of the jaw.

Moreover, the multipotentiality demonstrated by the postmortem-isolated ADSCs may give the rationale to hypothesize that these cells could also retain the capacity to differentiate into other cell types, such as smooth muscle cells or cardiomyocytes. In this sense, the translation of this study would be dramatically magnified as it would open up a brand new opportunity for obtaining cell models to understand the underlying mechanisms of lethal cardiomyopathies once the proband is already dead and an autopsy is performed. The direct impact of family screening and the global knowledge in these rare diseases warrant further efforts to explore the possibility of obtaining cardiomyocytes from postmortem ADSCs, which is now part of our future priorities.

Finally, the ethical concerns that arise when obtaining and manipulating pluripotential stem cells are not an issue when dealing with other sources of progenitors, such as MSCs or ADSCs, due to the fact that they cannot be used in human cloning [37]. In turn, using these cell types as therapeutics may promote undesired tumor growth [37,38], a concern to be addressed by ongoing clinical trials. Finally, the use of anonymized postmortem samples represents a step up, both as a promising source of new effective treatments in regenerative medicine and also as the last possibility to obtain unique experimental models in life-threatening genetic rare diseases. Aiming to improve research and future treatments, Ethic Committees can guarantee agreement with the Law in certain scenarios. Social support of organ donation from deceased donors generated in the past the concept of ‘presumed consent’ which, somehow, underlies the possible use of postmortem engineered tissues for these health-related topics.

## 4. Materials and Methods

### 4.1. Patient Enrolment

This study was conducted in accordance with the ethical standards of the Declaration of Helsinki. All individuals included in the study were dead at the moment of enrollment and the samples were obtained during the required forensic protocol mandatory by Spanish Law in cases of out-of-hospital unexpected deaths. This kind of death dramatically impacts on relatives’ mental health. For these reasons, no written informed consent was neither required nor obtained and experimental procedures included in this study were approved by our local Ethics Committee (#2018/0417 and 2020-411-1).

### 4.2. Samples Origin

One sample of subcutaneous adipose tissue taken with an 8 mm punch from eight cadaveric donors were obtained during the autopsy protocol at the Instituto de Medicina Legal y Ciencias Forenses de Valencia. Postmortem intervals of the samples included in this study ranged from 30 to 55 h. Abdominal subcutaneous adipose tissue samples were rinsed in sterile PBS and stored at 4 °C until their study within the first 24 h after tissue collection.

We also included as a reference group commercial ADSCs obtained from the Lonza company (PT-5006; Lonza, Basel, Switzerland).

### 4.3. ADSC Isolation and Characterization

Briefly, ADSCs were isolated as follows. The adipose tissue underwent collagenase digestion (25 mg/mL in high glucose DMEM medium) (Merck KGaA, Alemania, C9891, Thermo Fisher Scientific, Carlsband, CA, USA, EE.UU, 10938-025) and subsequent incubation in a 37 °C bath, shaking the suspension every 15 min. Phases were separated by decantation and the aqueous phase was filtered twice consecutively through a 100 µm and a 70 µm pore size filter. The final volume obtained was centrifuged at 190 g for 10 min at 20 °C to obtain a cell pellet containing the ADSCs. The supernatant was then removed, and the cell pellet was resuspended in 1 mL of ADSCs proliferation culture medium consisting in high glucose DMEM supplemented with 1% antibiotics (Pen Strep, Sigma-Merck KGaA, Darmstadt, Germany, Europe, 15140-122), 1% antifungals (Amphotericin B 100×X, EUROCLONE, Italia, Europe, ECM0009D), 2 mM L-Glutamine (Sigma-Merck KGaA, Darmstadt, Germany, Europe, 25030-081) and 10% inactivated FBS (Sigma-Merck KGaA, Darmstadt, Germany, Europe, 10270-106).

It was next seeded in T25 culture flasks and maintained in a humidified atmosphere containing 5% CO_2_ and 95% air at 37 °C. Medium was replaced every two days until confluence. Once expanded, cells were trypsinized into a single-cell suspension with 0.25% trypsin-EDTA (Sigma-Merck KGaA, Alemania, 25200-056). In total, 100,000 cells were characterized using flow cytometry in order to determine the culture purity of the mesenchymal cells, as previously reported [31]. To this end, a FACSCalibur equipped with a 488 nm argon laser and a 635 nm red diode laser (Becton Dickinson, Madrid, Spain) was used. The experimental data were analyzed with the CellQuest software version 5.1 (Becton Dickinson, Madrid, Spain). To exclude cellular debris, samples were sorted based on their light-scattering properties in the side-scattered and forward-scattered light modes, and 10,000 events per sample were collected within this gate (R1), using the medium setting for the sample flow rate.

Cells were characterized using flow cytometry based on the following markers: CD105 (Biolegend, San Diego, CA, USA, 323219), CD146 (Biolegend, San Diego, CA, USA, 361005), CD29 (Biolegend, San Diego, CA, USA, 303015), CD31 (Biolegend, San Diego, CA, USA, 303117), CD44 (Biolegend, San Diego, CA, USA, 103009), CD73 (R&D systems Inc., Minneapolis, MI, USA, FAB5795A), CD45 (Milteny, Germany, Europe, 130-110-769), and STRO-1 (Biolegend, San Diego, CA, USA, 340103).

### 4.4. ADSC Proliferation and Differentiation

Once characterized, ADSCs were cultivated in a proliferation medium until reaching a confluence of 60% for chondrocyte or osteocyte differentiation. Then, the culture medium was replaced with specific differentiation medium and further cultured for 4 weeks in case of pursuing chondrocyte differentiation or 2 weeks for osteocyte differentiation.

The chondral differentiation medium consisted of a high glucose DMEM medium enriched with 1% insulin-transferrin-sodium selenite medium supplement (Sigma-Merck KGaA, Germany, 41400–045), 50 µg/mL ascorbic acid (Sigma-Merck KGaA, Germany, A4403), 10 ng/mL TGF-β1 (Sigma-Merck KGaA, Germany, T7039), and 1% heat-inactivated FBS, as previously reported [25]. The osteogenic induction medium composition included αMEM (Sigma-Merck KGaA, Germany, 22571-020) supplemented with 10% FBS (Sigma-Merck KGaA, Germany, 10270-106) 1% P/S antibiotic solution, 2 mM L-glutamine, 10 μg/mL glycerol-2-phosphate (Sigma-Merck KGaA, Germany, G9422), 4 μg/mL ascorbic acid, 0.1 μg/mL dexamethasone (Sigma-Merck KGaA, Germany, D4902), and 50 ng/mL of BMP2 (Stemcell Technologies; France, 78004) [34].

### 4.5. Multiparametric Characterization of the Final Differentiated Cultures

Once the differentiation protocol was finished, the resulting cultures were characterized according to their morphological phenotype and protein expression profile. Phase contrast microscopy was used to assess cell arrangement, cell shape, and cytoplasmic content. Cytoplasm maturation in terms of F-actin expression was evaluated using fluorescence as follows. F-actin was evaluated using rhodamine-conjugated phalloidin (Molecular Probes, Thermo Fisher Scientific, Madrid, Spain, R415). Cells were washed with PBS and fixed in a 3.7% solution of formaldehyde in PBS for 10 min at room temperature. Next, they were permeabilized with 0.1% Triton X-100 in PBS for 3–5 min and blocked with PBS containing 1% BSA for 20–30 min to reduce non-specific background staining. Each sample was then stained for 20 min with 5 µL phalloidin methanol stock solution diluted in 200 µL PBS. Finally, after washing the samples with PBS, the nuclei were stained with DAPI (Sigma-Merck KGaA, Germany, 124653). Fluorescence images were captured with a Leica DM2500 fluorescence microscope (Leica, Madrid, Spain).

### 4.6. Determination of Main Markers of Each Cell Type

A dual approach was employed to evaluate the protein expression pattern in our cultures, firstly with protein immunofluorescent labelling and secondly with gene expression quantification by means of reverse transcriptase quantitative polymerase chain reaction (RT-qPCR).

Representative proteins for each desired cell type were selected. Type I and II collagen as well as aggrecan expression were determined to study chondrocyte differentiation. Type I collagen, osteocalcin, and phosphatase alkaline for osteocyte differentiation. Thus, our target genes were *COL2A1*, *COL1A1*, *ACAN*, *BGLAP*, and *ALPL*, depending on the cell type under evaluation.

#### 4.6.1. Specific Proteins of Each Cell Type Using Immunofluorescence

ADSCs were cultured in 8-well millicells (Sigma-Merck KGaA, Germany, PEZGS0816). Once confluent, cells were washed three times with cold PBS and fixed with 4% formalin for 10 min and permeabilized using 0.1% Triton X100 (Sigma-Merck KGaA, Germany, 112298) in PBS for 5 min at room temperature. After being blocked with 1% BSA (Sigma-Merck KGaA, Germany, A2153) solution in PBS to avoid non-specific binding, cells were incubated with the selected primary antibodies prepared in EnVision diluent (Dako, Denmark, K8006). For type I collagen and aggrecan (Sigma-Merck KGaA, Germany, C2456 and Santa Cruz Biotechnology, USA, sc-25674, respectively) the optimal dilution was 1:100, whereas for type II collagen (Sigma-Merck KGaA, Germany, CP18) was 1:500. Preparations were incubated overnight in darkness at 4 °C and in a wet chamber in order to prevent samples from drying out. Then, wells were washed with PBS three times and incubated with the secondary antibody. For all of them, a monoclonal anti-mouse secondary antibody bound to FITC was used at 1:200 (Sigma-Merck KGaA, Germany, F2883). Finally, wells were washed twice with PBS and slides were covered with mounting medium containing DAPI to stain the nuclei of the cells for 1 h, sealing the slides with nail varnish. Fluorescence images were captured with a Leica DM2500 fluorescence microscope (Leica, Madrid, Spain).

#### 4.6.2. Protocol to Quantify the Gene Expression of the Main Markers for Each Cell Type Using RT-qPCR

Total RNA was extracted from cultures using Trizol reagent (Thermo Fischer Scientific Inc., Waltham, MA, USA) according to the manufacturer’s instructions. RNA concentration was determined using spectrophotometry with a Nanodrop 2000 spectrophotometer (Fischer Scientific, Madrid, Spain). Only extracts with a ratio 260/280 > 1.8 were used. RNA integrity was evaluated by capillary electrophoresis using a Bioanalyzer (Agilent Technologies, Santa Clara, CA, USA). For the determination of gene expression levels, only extracts with an RNA Integrity Number (RIN) of ~10 were used. Random hexamers were used to synthesize complementary DNA (cDNA) using TaqMan RT reagents (Applied Biosystems, Foster City, CA, USA, N8080234) following the manufacturer’s instructions. Gene expression levels were assayed using reverse transcriptase quantitative polymerase chain reaction (RT-qPCR) using on Demand Assays (Applied Biosystems, Madrid, Spain). The following assays were used: *COL1A1* (Hs00164004_m1), *COL2A1* (Hs00264051_m1), *ACAN* (Hs00153936_m1), *BGLAP* (Hs01587814_g1) and *ALP* (Hs01029144_m1). Reactions were carried out in a 7900 HT real-time Thermocycler (Applied Biosystems, Madrid, Spain). The comparative ΔΔCt method with glyceraldehyde 3-phosphate dehydrogenase *(GAPDH*) was used as an endogenous control to calculate relative gene expression levels [39], which expresses the fold change in the experimental group with respect to the control group.

### 4.7. Data Presentation and Analyses

All the differentiations were carried out in triplicate. For microscopy experiments, representative images of 5 fields of each stain are presented. For relative gene expression analyses, data are presented as the mean ± SD. Statistical analyses were carried out using analysis of variance (ANOVA) followed by Tukey’s multiple comparison test (GraphPad Software version 5.0, Inc., San Diego, CA, USA). Significance was accepted at a *p*-level < 0.05.

## 5. Conclusions

In summary, our results demonstrate for the first time the feasibility of isolating viable ADSCs from subcutaneous adipose tissues from deceased human donors that can be further differentiated into chondrocytes and to osteocytes, even with postmortem intervals ranging from 30 to 55 h.

## Figures and Tables

**Figure 1 ijms-24-15696-f001:**
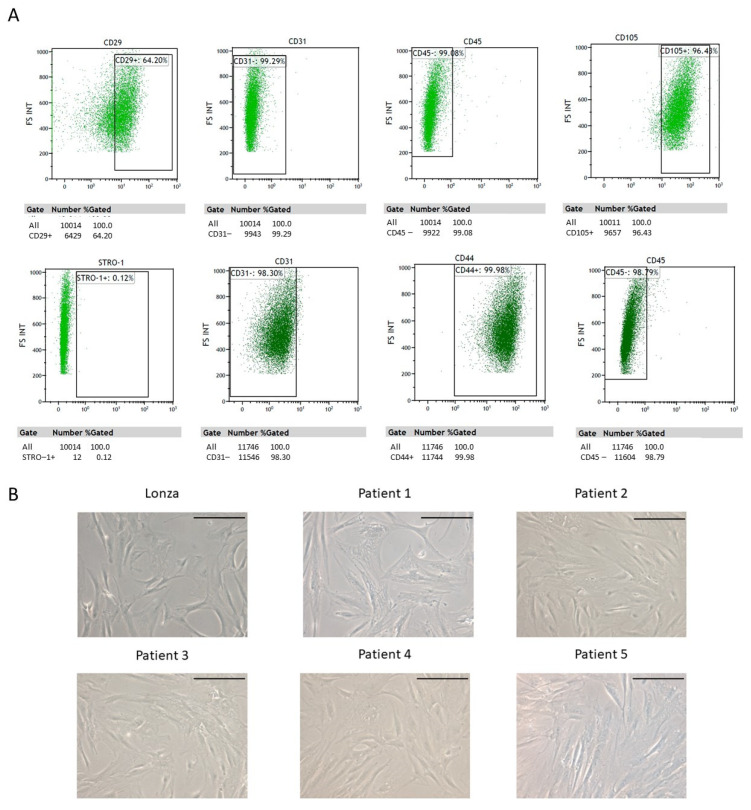
Adipose-derived stem cell (ADSC) characterization 1 week after being isolated and expanded in proliferation culture medium. (**A**) Results from flow cytometry of one of the analyzed samples. (**B**) Phase contrast microcopy images. A commercial cell line (Lonza) was used as reference. Scale bar equals to 200 µm.

**Figure 2 ijms-24-15696-f002:**
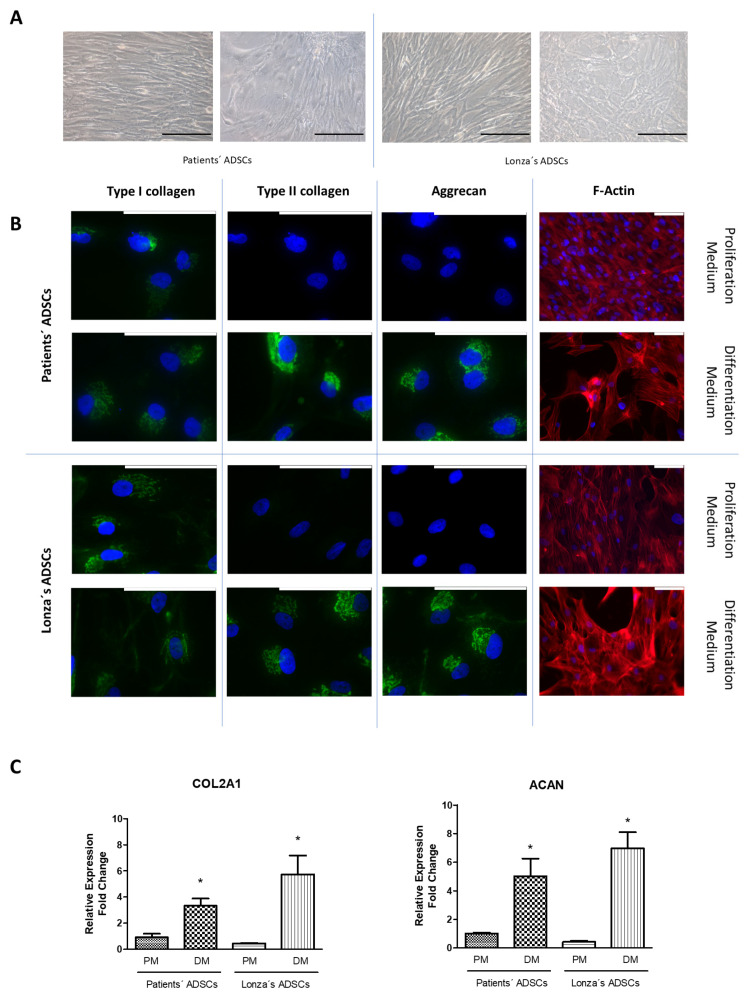
Chondrogenic differentiation of adipose-derived stem cells (ADSCs). Representative images of ADSCs from autopsies and from Lonza after having been cultured in chondrogenic medium for 4 weeks. (**A**) Morphological evaluation of cells was performed using phase contrast microscopy. (**B**) Type I and type II collagen and aggrecan protein expression were assessed using immunofluorescence. The distribution of the cytoskeleton was evaluated with the fluorescent staining of F-actin using phalloidin-rhodamine. (**C**) Relative gene expression of *COL2A1* and *ACAN* was estimated using real-time RT-qPCR. All experiments were performed in triplicate. Pictures are representative of 5 different fields. The patients’ series included the 8 samples that fulfilled the isolation and characterization protocol. Scale bar equals to 200 µm (in (**A**)) or 100 µm (in (**B**)). Gene expression data are represented as fold change, mean ± SD. * *p* < 0.05 when DM was compared to PM within each group. PM: proliferation medium. DM: differentiation medium.

**Figure 3 ijms-24-15696-f003:**
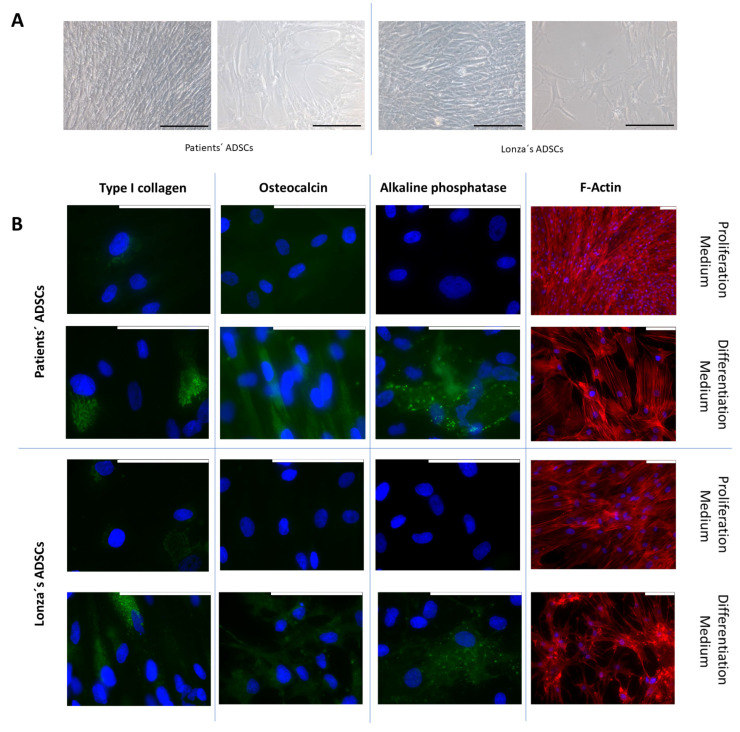
Osteogenic differentiation of adipose-derived stem cells (ADSCs). Representative ADSCs from autopsies and from Lonza after having been cultured in osteogenic medium for 2 weeks. (**A**) Morphological evaluation of cells was performed using phase contrast microscopy. (**B**) Type I collagen, as well as osteocalcin and alkaline phosphatase protein expression, was assessed using immunofluorescence. The distribution of the cytoskeleton was evaluated with the fluorescent staining of F-actin using phalloidin-rhodamine. (**C**) Relative gene expression of *COL1A1*, *BGLAP*, and *ALPL* was estimated using real-time RT-qPCR. All experiments were performed in triplicate. The patients’ series included the 8 samples that fulfilled the isolation and characterization protocol. Pictures are representative of 5 different fields. Scale bar equals to 200 µm (in (**A**)) or 100 µm (in (**B**)). Gene expression data are represented as fold change, mean ± SD. * *p* < 0.05 when DM was compared to PM within each group. PM: proliferation medium. DM: differentiation medium.

## Data Availability

Not applicable.

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
