# Peer review of "Cadaveric Adipose-Derived Stem Cells for Regenerative Medicine and Research"

_ijms, 2023, doi:10.3390/ijms242115696_

Round 1

Reviewer 1 Report

In the present investigation, authors aimed to study the cadaveric ADSCs as a cell source for regenerative medicine applications. The article is well organized and the results are well presented. However, the following are important minor issues to address:

1. In the abstract, there is no need to number the sentences. Please check and rewrite the abstract in an easier way for the readers.

2. In the result section, no need for introduction and referencing (For example Lines 120-122, and lines 153-156). check, mentioning your findings only are enough in this section. 

3. Line 167: do you mean Figure 3C instead of Figure 4C?

4. Some important publications in this area of study are missed in this manuscript. For example: https://doi.org/10.3390/ijms24087513

5. The English language needs to be checked to address the grammatical and typing errors.

6. It is better to write the conclusion of the present investigation in a separate section to be easier for the readers,

7. In section 4.2 Samples origin, from which area did you collect the subcutaneous tissue samples?

The English language needs to be checked to address the grammatical and typing errors.

Author Response

As the corresponding author and on behalf of all the authors I want, on the one hand, to thank the editor for the opportunity to respond to the reviewers' comments. On the other hand, I would like to sincerely thank reviewers 1 and 2 for their support and help reviewing this manuscript. We have submitted a new version of the manuscript including the changes suggested by the reviewers (highlighted).

Reviewer 1

In the present investigation, authors aimed to study the cadaveric ADSCs as a cell source for regenerative medicine applications. The article is well organized and the results are well presented. However, the following are important minor issues to address:

Comment: In the abstract, there is no need to number the sentences. Please check and rewrite the abstract in an easier way for the readers.

Reply: We sincerely appreciate the reviewer comment. The abstract has been rewritten according to reviewer instructions.

Comment: In the result section, no need for introduction and referencing (For example Lines 120-122, and lines 153-156). check, mentioning your findings only are enough in this section.

Reply: We appreciate the reviewer's appreciation on this point. However, we believe that these references in the results section are necessary to contextualize the findings presented in this work, especially taking into account the enormous heterogeneity in protocols related to the differentiation of chondrocytes or osteocytes.

Comment: Line 167: do you mean Figure 3C instead of Figure 4C?

Reply: We apologize to the reviewer for this error, which has been corrected in the new version of the manuscript.

Comment: Some important publications in this area of study are missed in this manuscript. For example: https://doi.org/10.3390/ijms24087513.

Reply: We sincerely appreciate the reviewer's suggestion in this regard. The suggested reference corresponds to a study that evaluates the impact of adipose tissue depot harvesting site on the multilineage induction capacity of male rat adipose-derived mesenchymal stem cells. In this study, we have not studied this aspect, since we have not compared the potential of the isolated ADSCs depending on the location of the sample, so we understand that it is not appropriate to include it in the revised version of the manuscript. However, we will take it into account in future research.

Comment: The English language needs to be checked to address the grammatical and typing errors.

Reply: We thank the reviewer again for this comment with which we completely agree. The manuscript has been reviewed by a native English person. We have included an acknowledgment section thanking for her support.

Comment: It is better to write the conclusion of the present investigation in a separate section to be easier for the readers.

Reply: We have included this section in the revised version of the manuscript following the reviewer's instructions.

Comment: In section 4.2 Samples origin, from which area did you collect the subcutaneous tissue samples?

Reply: From subcutaneous abdominal adipose tissue. This information has been included in the section 4.2

Reviewer 2 Report

Consider  Zizhen et al., I find it a capital work on this topic. Also, I find an issue with a term  "adipose stem cells". The term usually used is "adipose-derived stem cells". In text, this term is used in that form.  In fig. 1 [captions] abbreviation "ADSC" should be gi ven in full first time it is used. Same is with caption of fig. 2.

Again, My primary concern is the p-value given as <0.05; besides that, I have more than a few additional concerns.

1. Please DO NOT use the abbreviation as the first word of the sentence. I refer to "adipose stem cell, abbreviated to "ADSC" in the ln. 47, and is used as the abbreviation after that. Also, using an abbreviation at the beginning of the figure caption is not fitting.
2. Main text, figure legends, or abstract are all separate and independent parts of a MS. So, wherever you use an abbreviation first in each part, the abbreviation must be given in full.
3. Authors are referred to the "instructions for authors"; "The abstract should be a total of about 200 words maximum. The abstract should be a single paragraph and should follow the style of structured abstracts, BUT WITHOUT HEADINGS…" including numerals ( 1)…). See the articles published.
4. Your references are outdated; at least 50% of references should have been published in the past 3 yrs.

Author Response

As the corresponding author and on behalf of all the authors I want, on the one hand, to thank the editor for the opportunity to respond to the reviewers' comments. On the other hand, I would like to sincerely thank reviewers 1 and 2 for their support and help reviewing this manuscript. We have submitted a new version of the manuscript including the changes suggested by the reviewers (highlighted).

Reviewer 2

Comment: Consider  Zizhen et al., I find it a capital work on this topic. Also, I find an issue with a term  "adipose stem cells". The term usually used is "adipose-derived stem cells". In text, this term is used in that form.  In fig. 1 [captions] abbreviation "ADSC" should be given in full first time it is used. Same is with caption of fig. 2.

Reply: We appreciate the reviewer's comment. In the new version of the manuscript the term adipose stem cells has been replaced by adipose-derived stem cells. We have also modified the figure 1 and 2 captions following the reviewer's recommendations.

Comment: Again, My primary concern is the p-value given as <0.05; besides that, I have more than a few additional concerns.

Reply: Once again we appreciate the reviewer's comment. In this study our purpose was not to study whether there were significant differences between patients, but rather whether the isolated cells could be differentiated into chondrocytes or osteocytes. For this reason, we have chosen this p value, used extensively in in vitro studies or with laboratory animals.

Comment: Please DO NOT use the abbreviation as the first word of the sentence. I refer to "adipose stem cell, abbreviated to "ADSC" in the ln. 47, and is used as the abbreviation after that. Also, using an abbreviation at the beginning of the figure caption is not fitting.

Reply: We have modified the manuscript based on the reviewer's suggestion.

Comment: Main text, figure legends, or abstract are all separate and independent parts of a MS. So, wherever you use an abbreviation first in each part, the abbreviation must be given in full.

Reply: We thank the reviewer for this comment. We have modified the manuscript following his instructions.

Comment: Authors are referred to the "instructions for authors"; "The abstract should be a total of about 200 words maximum. The abstract should be a single paragraph and should follow the style of structured abstracts, BUT WITHOUT HEADINGS…" including numerals ( 1)…). See the articles published.

Reply: Once again we appreciate the reviewer's comment. In the new version of the manuscript the abstract has been re-written following his instructions.

Comment: Your references are outdated; at least 50% of references should have been published in the past 3 yrs.

Reply: Again, we appreciate the reviewer's comment. We want to clarify that the evidence published in relation to the potential of ADSCs, their ability to differentiate into chondrocytes or osteocytes and their potential in tissue engineering is not recent and was described many years ago. On the other hand, the literature regarding the use of post-mortem tissues for the isolation of MSCs is very limited. This is why the references included in the introduction of this manuscript are not very recent. However, the references included in the review are more current.